# THE ROBUSTNESS PLATEAU: WHY SKIP WEIGHT TUNING IS UNNECESSARY IN BATCH-NORMALIZED RESNETS

## ABSTRACT

Mean Field Theory predicts that deep residual networks require skip connection weights $\alpha = 1/\sqrt{L}$ (where $L$ is depth) to achieve optimal gradient flow. We empirically test this prediction on Batch-Normalized Pre-Activation ResNets trained on CIFAR-10. Our experiments across 57 configurations reveal that the $1/\sqrt{L}$ formula has no predictive power for BN-ResNets ($R^2 = 0.13$). Instead, we discover a *robustness plateau*: performance remains remarkably stable ($\sigma < 0.7\%$) across a $10\times$ range of skip weights ($\alpha \in [0.5, 2.0]$). We explain this discrepancy by noting that Mean Field Theory derivations assume networks *without* normalization layers. Batch Normalization already solves the gradient stability problem that $1/\sqrt{L}$ scaling addresses, rendering skip weight optimization unnecessary in practical BN-ResNets. Our findings suggest practitioners need not tune skip weights when using batch normalization.

## 1 INTRODUCTION

Residual networks (He et al., 2016a) have become foundational in deep learning, enabling training of networks with hundreds of layers through skip connections. The standard residual block computes:

$$h_{l+1} = h_l + \alpha \cdot F(h_l; W_l) \tag{1}$$

where $\alpha$ is the skip weight (typically $\alpha = 1$) and $F$ represents the residual branch.

Recent theoretical work using Mean Field Theory has derived optimal skip weights for deep ResNets. Yang & Schoenholz (2017) showed that skip connections enable ResNets to operate at the "edge of chaos," with subexponential rather than exponential gradient dynamics. Arpit et al. (2019) proposed $\alpha = 1/\sqrt{L}$ scaling based on mean field approximation to prevent vanishing/exploding gradients. Zhang et al. (2019) proved this scaling is mathematically sharp for forward stability, and Marion et al. (2025) showed it is the only non-trivial scaling in the large-depth limit.

However, these theoretical results share a critical assumption: **they analyze networks without normalization layers**. Virtually all practical ResNets use Batch Normalization (BN) (Ioffe & Szegedy, 2015), which may render the theoretical predictions inapplicable.

In this work, we empirically test Mean Field Theory predictions on BN-ResNets. We find that: (1) the $1/\sqrt{L}$ formula has no predictive power; (2) a robustness plateau exists where skip weights $\alpha \in [0.5, 2.0]$ yield nearly identical performance; and (3) this is explained by BN already providing the gradient stability that skip weight scaling addresses.

## 2 METHODS

**Architecture.** We use Pre-Activation ResNets (He et al., 2016b) with Batch Normalization, trained on CIFAR-10. We scale the residual branch (not identity): $h_{l+1} = h_l + \alpha \cdot F(h_l)$.

**Experiments.** We conduct 57 experiments across three phases: (1) Skip weight sweep with $\alpha \in [0.1, 2.0]$ and depths $L \in \{9, 15, 27, 54\}$; (2) Validation at new depths $L \in \{21, 42\}$ with

theory-predicted $\alpha = 1/\sqrt{L}$; (3) Depth-varying schedules including linear, U-shaped, and learned (LayerScale) weights.

**Training.** All models train for 200 epochs with SGD (momentum 0.9), learning rate 0.1 (reduced $10\times$ at epochs 100, 150), weight decay $5 \times 10^{-4}$, and standard augmentation.

## 3 RESULTS

### 3.1 MEAN FIELD THEORY DOES NOT PREDICT OPTIMAL SKIP WEIGHTS

Figure 1 shows empirical optimal skip weights versus the theoretical $1/\sqrt{L}$ prediction. The theory predicts $\alpha$ should decrease from 0.33 (at $L$=9) to 0.14 (at $L$=54). Instead, empirical optima show no consistent pattern with depth.

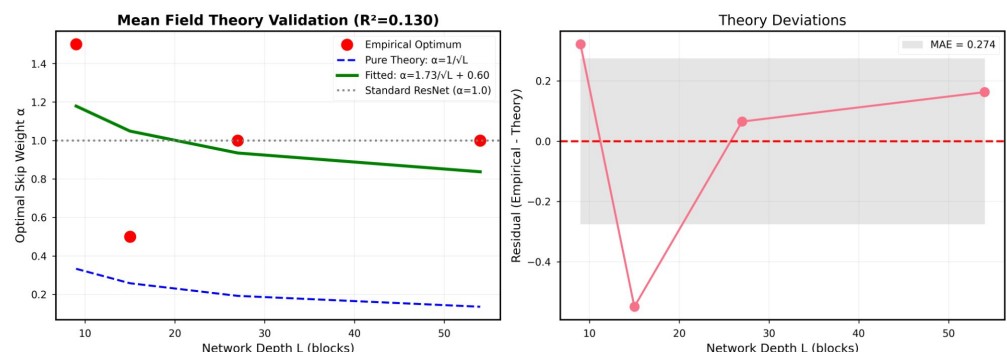

Figure 1: **Mean Field Theory validation fails for BN-ResNets.** Left: Empirical optimal $\alpha$ (red dots) versus theoretical prediction $\alpha = 1/\sqrt{L}$ (blue dashed). The fitted curve (green) shows a weak relationship ($R^2 = 0.13$). Right: Residuals show no systematic pattern. The standard $\alpha = 1.0$ (gray dotted) performs well across all depths.

Quantitatively, the correlation between predicted and empirical optima is weak ($r = 0.46$) and the $R^2 = 0.13$ indicates the theory explains only 13% of variance. The Mean Absolute Error of 0.27 is substantial given the predicted range of 0.14–0.33.

### 3.2 THE ROBUSTNESS PLATEAU

Rather than a sharp optimum at $\alpha = 1/\sqrt{L}$, we observe a broad plateau of near-optimal performance. Figure 2 shows the phase diagram across skip weights and depths.

Within $\alpha \in [0.5, 2.0]$ (27 experiments), accuracy is remarkably stable: mean 93.87%, std 0.68%. This represents a **$10\times$ range in skip weight with $<2\%$ accuracy variation**. Table 1 shows representative results.

Table 1: Robustness plateau: accuracy varies minimally across skip weights ($L$=27).

| $\alpha$ | 0.25 | 0.50 | 0.75 | 1.00 | 1.25 | 1.50 | 2.00 |
|---|---|---|---|---|---|---|---|
| Acc (%) | 93.75 | 94.06 | 94.34 | **94.45** | 94.17 | 94.21 | 94.00 |

### 3.3 GRADIENT FLOW ANALYSIS

Figure 3 shows gradient magnitude profiles across layers. Plain networks ($\alpha = 0$) exhibit severe gradient explosion at depth, spanning 5+ orders of magnitude. In contrast, all ResNets with $\alpha > 0$ show remarkably similar, bounded gradient profiles regardless of skip weight value.

This clustering explains the robustness plateau: BN normalizes activations to unit variance at each layer, absorbing the variance changes that would otherwise require careful $\alpha$ tuning.

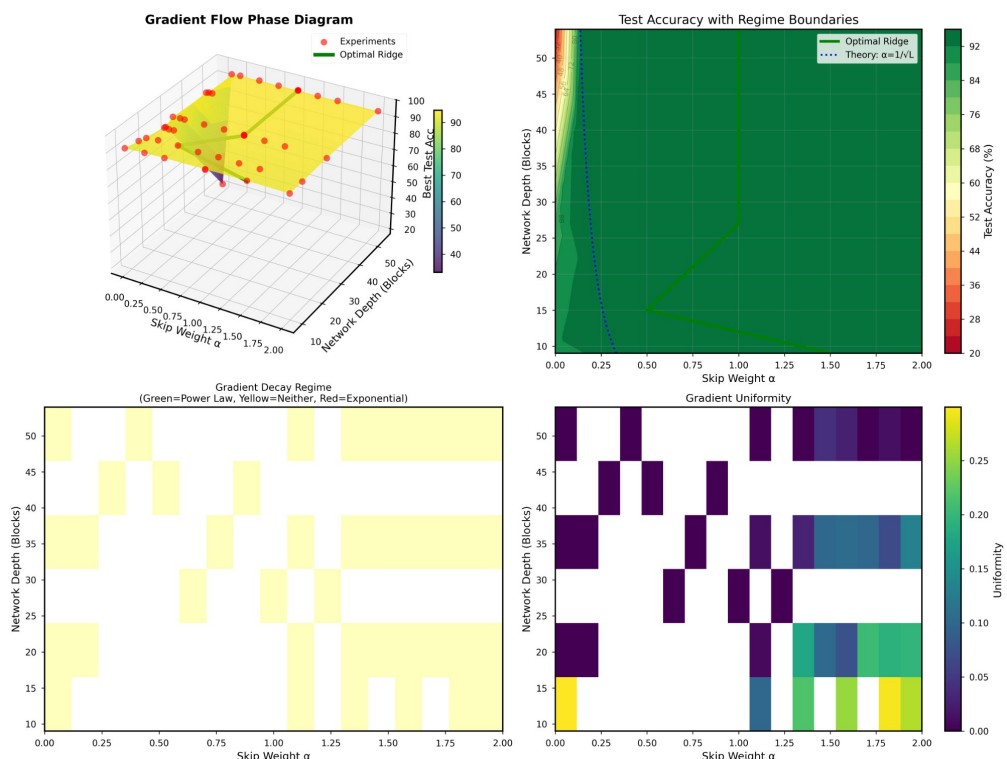

Figure 2: **Phase diagram reveals robustness plateau.** Top-left: 3D surface showing accuracy across $(\alpha, L)$ space. Top-right: 2D heatmap with theoretical $1/\sqrt{L}$ curve (blue dashed) and empirical optimal ridge (green). The theory curve lies far from the high-accuracy region for deeper networks. Bottom: All experiments show "neither" decay type (yellow), indicating BN prevents clean exponential/power-law gradient decay.

### 3.4 SKIP WEIGHT SCHEDULES PROVIDE NO BENEFIT

We tested depth-varying schedules including theory-motivated $1/\sqrt{L}$ (B1), linear increase/decrease (S1/S2), U-shaped (S3), exponential decay (S4), inverse depth (S5), and learned LayerScale (B3). Figure 4 shows results.

The standard uniform $\alpha = 1.0$ (B2) achieves the best performance (94.66%), outperforming both the theory-motivated $1/\sqrt{L}$ schedule (B1: 94.09%) and learned weights (B3: 94.43%). Complex schedules provide no benefit over the default.

## 4 DISCUSSION

**Why Mean Field Theory doesn't apply to BN-ResNets.** The theoretical derivations analyze gradient flow *at initialization* in networks *without normalization*. Without BN, activation variance grows as $(1 + \alpha^2)^L$, requiring $\alpha = O(1/\sqrt{L})$ to prevent explosion. With BN, each layer's output is explicitly normalized to unit variance, eliminating this dependency.

**BN and skip weight scaling are redundant mechanisms.** Both solve the same problem (gradient stability) through different means. When BN is present, the careful $\alpha$ tuning becomes unnecessary. This explains why the standard $\alpha = 1.0$ works across all depths.

**Practical implications.** Practitioners using BN-ResNets need not tune skip weights. The default $\alpha = 1.0$ is near-optimal, and any value in $[0.5, 2.0]$ yields similar performance. Resources are better spent on other hyperparameters.

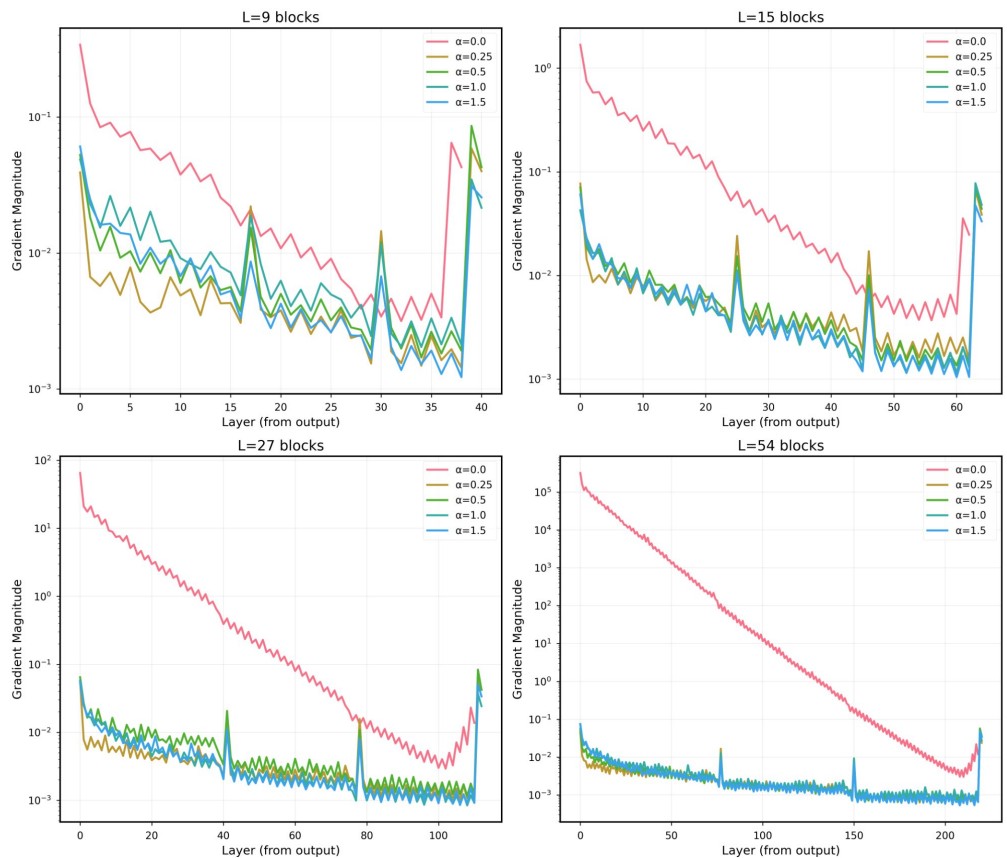

Figure 3: **BN-ResNets show skip-weight-invariant gradient profiles.** Gradient magnitudes vs. layer for different depths. Pink ($\alpha$=0, plain networks) shows catastrophic explosion at depth. All ResNets ($\alpha > 0$) cluster tightly regardless of skip weight, demonstrating BN's stabilizing effect.

**Theoretical implications.** Mean Field Theory correctly identifies that gradient stability requires *either* $\alpha = 1/\sqrt{L}$ scaling *or* normalization layers. Our results validate this disjunction empirically: with BN providing stability, the skip weight constraint relaxes dramatically.

## 5 CONCLUSION

We empirically tested Mean Field Theory predictions for skip connection weights in ResNets. The $1/\sqrt{L}$ formula, derived for un-normalized networks, has no predictive power for practical BN-ResNets. Instead, we discover a robustness plateau where performance is insensitive to skip weight choice. This finding clarifies the scope of existing theory and provides practical guidance: when using batch normalization, skip weight tuning is unnecessary.

ACKNOWLEDGMENTS

Acknowledgments will be added in the camera-ready version.

REFERENCES

Devansh Arpit, Víctor Campos, and Yoshua Bengio. How to initialize your network? robust initialization for weightnorm & resnets. In *Advances in Neural Information Processing Systems*, volume 32, 2019.

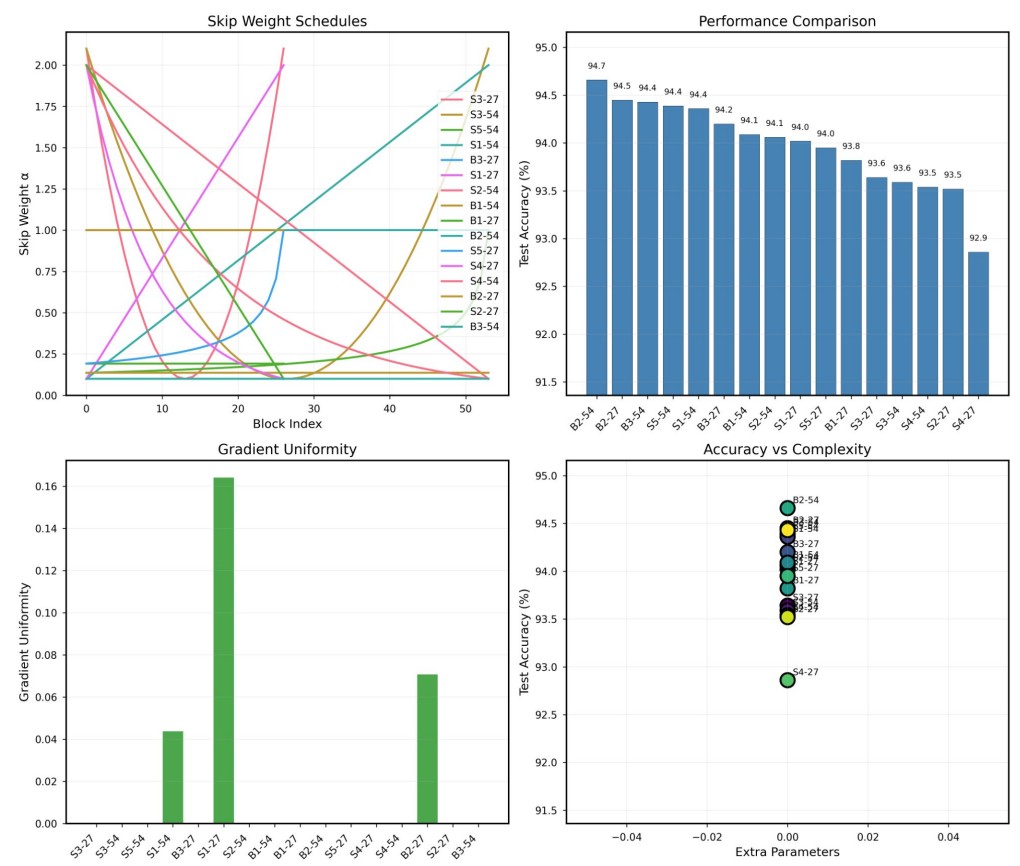

Figure 4: **No schedule outperforms uniform α=1.0.** Top-left: Schedule shapes. Top-right: Uniform α=1.0 (B2) achieves best accuracy. The theory-motivated B1 ($\alpha = 1/\sqrt{L}$) underperforms standard ResNets.

Kaiming He, Xiangyu Zhang, Shaoqing Ren, and Jian Sun. Deep residual learning for image recognition. In *Proceedings of the IEEE Conference on Computer Vision and Pattern Recognition*, pp. 770–778, 2016a.

Kaiming He, Xiangyu Zhang, Shaoqing Ren, and Jian Sun. Identity mappings in deep residual networks. In *European Conference on Computer Vision*, pp. 630–645. Springer, 2016b.

Sergey Ioffe and Christian Szegedy. Batch normalization: Accelerating deep network training by reducing internal covariate shift. In *International Conference on Machine Learning*, pp. 448–456. PMLR, 2015.

Pierre Marion, Raphaël Berthier, Florent Krzakala, et al. Scaling resnets in the large-depth regime. *Journal of Machine Learning Research*, 26:1–48, 2025.

Greg Yang and Samuel S Schoenholz. Mean field residual networks: On the edge of chaos. *Advances in Neural Information Processing Systems*, 30, 2017.

Huishuai Zhang, Wei Da, and Chen Wei. Stabilize deep resnet with a sharp scaling factor $\tau$. *arXiv preprint arXiv:1903.07120*, 2019.

# A APPENDIX: EXTENDED RESULTS

## A.1 COMPLETE EXPERIMENTAL RESULTS

Table 2 shows all uniform skip weight experiments.

Table 2: Complete results for uniform skip weight experiments.

| Depth | $\alpha$ | Accuracy (%) | Gradient SNR | Theory $\alpha$ |
|---|---|---|---|---|
| 9 | 0.25 | 92.21 | 0.522 | 0.333 |
| 9 | 0.50 | 92.56 | 0.565 | 0.333 |
| 9 | 1.00 | 92.64 | 0.570 | 0.333 |
| 9 | 1.50 | 93.11 | 0.613 | 0.333 |
| 9 | 2.00 | 92.89 | 0.598 | 0.333 |
| 15 | 0.10 | 92.85 | 0.435 | 0.258 |
| 15 | 0.25 | 93.48 | 0.548 | 0.258 |
| 15 | 0.50 | 93.69 | 0.589 | 0.258 |
| 15 | 0.75 | 93.24 | 0.560 | 0.258 |
| 15 | 1.00 | 93.57 | 0.566 | 0.258 |
| 15 | 1.25 | 93.40 | 0.606 | 0.258 |
| 15 | 1.50 | 93.49 | 0.607 | 0.258 |
| 15 | 2.00 | 93.21 | 0.611 | 0.258 |
| 27 | 0.10 | 93.48 | 0.429 | 0.193 |
| 27 | 0.25 | 93.75 | 0.565 | 0.193 |
| 27 | 0.50 | 94.06 | 0.587 | 0.193 |
| 27 | 0.75 | 94.34 | 0.587 | 0.193 |
| 27 | 1.00 | 94.45 | 0.599 | 0.193 |
| 27 | 1.25 | 94.17 | 0.585 | 0.193 |
| 27 | 1.50 | 94.21 | 0.598 | 0.193 |
| 27 | 2.00 | 94.00 | 0.608 | 0.193 |
| 54 | 0.25 | 94.55 | 0.558 | 0.136 |
| 54 | 0.50 | 94.59 | 0.536 | 0.136 |
| 54 | 0.75 | 94.47 | 0.582 | 0.136 |
| 54 | 1.00 | 94.74 | 0.602 | 0.136 |
| 54 | 1.25 | 94.38 | 0.589 | 0.136 |
| 54 | 1.50 | 94.50 | 0.599 | 0.136 |
| 54 | 2.00 | 94.56 | 0.588 | 0.136 |

## A.2 VALIDATION EXPERIMENTS AT NEW DEPTHS

To ensure our findings generalize, we tested at depths not seen during the main sweep:

$L$=21 (theory predicts $\alpha = 0.218$): Accuracies were 93.68% ($\alpha$=0.16), 93.52% ($\alpha$=0.218), and 93.68% ($\alpha$=0.28). The theory-predicted value performed no better than neighbors.

$L$=42 (theory predicts $\alpha = 0.154$): Accuracies were 93.85% ($\alpha$=0.12), 94.34% ($\alpha$=0.154), and 94.39% ($\alpha$=0.20). Again, no advantage for the theory-predicted value.

## A.3 PLAIN NETWORK DEGRADATION

Plain networks without skip connections show the degradation problem that ResNets solve:

Table 3: Plain network degradation with depth.

| Network | Blocks | Accuracy (%) | Gradient SNR |
|---|---|---|---|
| P-20 | 9 | 91.62 | 0.625 |
| P-32 | 15 | 89.76 | 0.625 |
| P-56 | 27 | 86.12 | 0.408 |
| P-110 | 54 | 22.12 | 0.026 |

The 69.5 percentage point degradation from $L$=9 to $L$=54 confirms the vanishing gradient problem that skip connections address.

## CHALLENGE: SCIENCE OF DL IMPROVEMENT

### WHAT MODEL ARE YOU TARGETING?

We target **Deep Residual Networks (ResNets)** utilizing **Batch Normalization (BN)** and Pre-Activation blocks (He et al., 2016b). Historically, these models were designed to solve the "degradation problem" in very deep architectures (He et al., 2016a). While current literature using Mean Field Theory suggests that deep ResNets require strictly scaled skip connection weights ($\alpha = 1/\sqrt{L}$) to maintain stable gradient flow and avoid vanishing or exploding gradients at initialization (Arpit et al., 2019; Zhang et al., 2019; Marion et al., 2025), we target the gap between these theoretical un-normalized models and practical, normalized architectures.

### HOW DO YOUR RESULTS CONTRIBUTE TO UNDERSTANDING THESE MODELS?

Our work provides a scientific explanation for why the theoretical $1/\sqrt{L}$ scaling law fails to predict optimal performance in practical BN-ResNets ($R^2 = 0.13$). We identify that **Batch Normalization and skip weight scaling are redundant stability mechanisms**. Our gradient flow analysis (Figure 3) demonstrates that BN acts as a primary stabilizer by normalizing activation variance at every layer, which effectively "absorbs" the variance growth $(1 + \alpha^2)^L$ that the $1/\sqrt{L}$ scaling was designed to counteract. By identifying the **Robustness Plateau**, we show that the gradient stability constraint on skip weights is significantly more relaxed than previously understood, provided normalization is present.

### HOW DO YOU EXPECT YOUR SUBMISSION TO INFLUENCE FUTURE WORK?

These findings have two immediate impacts on the science of deep learning:

- **Architectural Simplicity:** We demonstrate that practitioners can rely on the default uniform skip weight ($\alpha = 1.0$) without performance loss, eliminating the need for hyperparameter tuning of skip weights or complex schedules like LayerScale in normalized networks.
- **Theoretic Refinement:** We challenge the community to develop unified theoretical frameworks that account for both skip connections and normalization layers simultaneously. Current Mean Field Theory is a sharp tool for un-normalized networks, but our work highlights the necessity for a "normalized theory" to better reflect the state of practical deep learning.

