# OpenReview forum: "The Robustness Plateau: Why Skip Weight Tuning is Unnecessary in Batch-Normalized ResNets"
_ICLR.cc/2026/Workshop/Sci4DL — Submitted to Sci4DL 2026_

### Official Review · Reviewer_TvoK · 2026-02-23

**Fit:** 2
**Significance:** 1
**Confidence:** 3

**Summary:**

This work presents empirical evidence for the needlessness of skip-weight scaling for ResNets, due to the stabilization effects of BatchNorm.

**Strengths:**

The paper explores a broad and carefully designed set of experiments, covering 57 configurations across different depths, skip weights, unseen validation depths, and skip-weight schedules. Importantly, it goes beyond reporting final accuracies by also analyzing gradient flow, comparing against plain networks, and testing alternative schedules, which makes the conclusions more convincing and less reliant on surface-level performance metrics.

**Suggestions:**

- I'm not sure how relevant the results are in the absence of 1. learning rate sweep, 2. learning rate schedules, 3. random repeats, etc. Especially 3. because it is unreasonable to draw any conclusions based on a comparison across one sample from each setup.

- Does the training converge in 200 epochs of training? In my experience, training ResNets on CIFAR-10 for 200 epochs isn't usually enough for convergence, especially using SGD. With regards to Figure 1, I would've liked to see the raw loss/accuracy values for different values of $\alpha$, just to get a sense of how far the theoretical predictions are from the empirical results. If the difference isn't large, the deviations could very well be because of the single sample used. *Theory is rarely perfect, but it's valuable to know if it's good enough.*

- Aren't Figures 2a and 2b the same? If so, it would make sense to remove one. I suggest 2a, since the dip in accuracy on the top right corner isn't really visible. Even the variations, albeit small, aren't really visible with most shades being yellow. If they are not the same, please help me understand the difference. And in this case, it would be good to use the same colormap for both figures. Also, in Figure 2, a better choice of colors could be made to improve visibility of the optimal ridge.

- What is "best test accuracy" in Figure 2a? Best over what? From what I understand the network depth and skip weights are controlled. What are the other axes of variation?

- I don't understand the point made in Section 3.3. You say that $\alpha=0$ exhibits gradient explosion, while all $\alpha>0$ show stability. If anything, doesn't that mean that residual connections, with whatever skip weight, solve the gradient explosion problem? How do you infer anything about BatchNorm from this? The normalization effect of BatchNorm on activations should also be present in the absence of residual connections, so $\alpha=0$ should've been stable as well, no?

- Could you help me understand why gradients with respect to the final layers explode with depth (near 0 in Fig. 3)? And what changes with residual connections to have such remarkable stability across models with different depths?

> The Mean Absolute Error of 0.27 is substantial given the predicted range of 0.14-0.33.

Do mention, in the main text, what the error refers to. Also, what range is being referred to here?

> This represents a 10× range in skip weight with <2% accuracy variation.

Why 10×? The text says you use $\alpha \in [0.5, 2.0]$. What does "accuracy variation" mean?

---

### Official Review · Reviewer_cWXt · 2026-02-26

**Fit:** 2
**Significance:** 1
**Confidence:** 3

**Summary:**

The paper empirically shows that the $1/\sqrt{L}$ scaling of the residual branch in ResNet architectures is unnecessary when Batch Normalization is used.

**Strengths:**

-

**Suggestions:**

Unfortunately, this work suffers from a severe lack of literature review. The main theme of the paper, namely, the effects of normalization layers in residual networks have been explored, both theoretically and empirically, in the following paper: https://arxiv.org/abs/2111.12143.
While the experiment with varying $\alpha$ values (figure 2) is new, the overall message of the paper is largely redundant. Consequently, it is difficult for me to suggest ways to improve the paper.
(This is the reason behind my harsh significance rating -- while the work is correct, its contribution is trivial in light of prior work.)

Also, note that "mean-field theory doesn't apply to BN-ResNets" is an incorrect take-away. A proper treatment of mean-field theory correctly captures the behaviour of ResNets with normalization layers.

---

### Official Review · Reviewer_vsbc · 2026-03-01

**Fit:** 3
**Significance:** 2
**Confidence:** 2

**Summary:**

This paper investigates the empirical validity of Mean Field Theory predictions for optimal skip connection weights (α) in deep residual networks with batch-normalization layers. While theory suggests that skip weights should scale as α=1/L​ (where L is depth) to maintain gradient stability , the authors demonstrate that this formula has virtually no predictive power for Batch-Normalized (BN) ResNets. Moreover, the authors show that performance remains remarkably stable across a 10x range of skip weights (α∈[0.5,2.0]).

**Strengths:**

1- The hypothesis is empirically validated on multiple experiments conducted on ResNet model with varied depths (L) and skip weights ($\alpha$). The results show no clear pattern in empirical optima (in Figure-1) and highlights "robustness plateau" (inTable-1).

2- The experimental findings shows that BN-ResNets do not need an additional exercise to tune skip-weights.

**Suggestions:**

The paper can be improved in following areas:

1- Extending experiments to more challenging datasets such as CIFAR-100 and Image Net.
2- The paper focuses specifically on Batch Normalization. It remains unclear if other normalization techniques (e.g., LayerNorm or GroupNorm) exhibit the same redundancy with skip scaling.
3- What will happen for architectures where normalization is applied after the residual addition rather than in the pre-activation style?

Additional comments:
1- I could not follow figure-2. It needs to be explained better in text emphasizing the experimental setup, x and y-axis description and clear findings/patterns.

---

### Meta-Review · Area_Chair_UYDJ · 2026-03-01

**Recommendation:** Reject

**Metareview:**

The paper lacks proper literature review that would help situate it with respect to previous work and highlight its novelty. In the absence of such review covering relevant work, it is difficult to see the novelty of this work. Several technical details are also missing or incomplete. I suggest the authors carefully read the reviewers' feedback and take it into account.

---

### Decision · Program_Chairs · 2026-03-02

Reject